# Joint M-Best-Diverse Labelings as a Parametric Submodular Minimization

**Alexander Kirillov**[1]    **Alexander Shekhovtsov**[2]    **Carsten Rother**[1]    **Bogdan Savchynskyy**[1]

[1] TU Dresden, Dresden, Germany        [2] TU Graz, Graz, Austria

`alexander.kirillov@tu-dresden.de`

## Abstract

We consider the problem of jointly inferring the $M$-best diverse labelings for a binary (high-order) submodular energy of a graphical model. Recently, it was shown that this problem can be solved to a global optimum, for many practically interesting diversity measures. It was noted that the labelings are, so-called, nested. This nestedness property also holds for labelings of a class of parametric submodular minimization problems, where different values of the global parameter $\gamma$ give rise to different solutions. The popular example of the parametric submodular minimization is the monotonic parametric max-flow problem, which is also widely used for computing multiple labelings. As the main contribution of this work we establish a close relationship between diversity with submodular energies and the parametric submodular minimization. In particular, the joint $M$-best diverse labelings can be obtained by running a *non-parametric* submodular minimization (in the special case - max-flow) solver for $M$ different values of $\gamma$ *in parallel*, for certain diversity measures. Importantly, the values for $\gamma$ can be computed in a closed form in advance, prior to any optimization. These theoretical results suggest two simple yet efficient algorithms for the joint $M$-best diverse problem, which outperform competitors in terms of runtime and quality of results. In particular, as we show in the paper, the new methods compute the exact $M$-best diverse labelings faster than a popular method of Batra et al., which in some sense only obtains approximate solutions.

## 1   Introduction

A variety of tasks in machine learning, computer vision and other disciplines can be formulated as energy minimization problems, also known as *Maximum-a-Posteriori* (MAP) or *Maximum Likelihood* (ML) estimation problems in undirected graphical models (Markov or Conditional Random Fields). The importance of this problem is well-recognized, which can be seen by the many specialized benchmarks [36, 21] and computational challenges [10] for its solvers. This motivates the task of finding *the most probable* solution. Recently, a slightly different task has gained popularity, both from a practical and theoretical perspective. The task is not only to find the most probable solution but multiple *diverse* solutions, all with low energy, see e.g., [4, 31, 22, 23]. The task is referred to as the "$M$-best-diverse problem" [4], and it has been used in a variety of scenarios, such as: (a) Expressing uncertainty of the computed solutions [33]; (b) Faster training of model parameters [16]; (c) Ranking of inference results [40]; (d) Empirical risk minimization [32]; (e) Loss-aware optimization [1]; (f) Using diverse proposals in a cascading framework [39, 35].

This project has received funding from the European Research Council (ERC) under the European Union's Horizon 2020 research and innovation programme (grant agreement No 647769). A. Shekhovtsov was supported by ERC starting grant agreement 640156.

In this work we build on the recently proposed formulation of [22] for the $M$-best-diverse problem. In this formulation all $M$ configurations are inferred *jointly*, contrary to the well-known method [4, 31], where a sequential, greedy procedure is used. Hence, we term it "joint $M$-best-diverse problem". As shown in [22, 23], the joint approach qualitatively outperforms the sequential approach [4, 31] in a number of applications. This is explained by the fact that the sequential method [4] can be considered as an approximate and greedy optimization technique for solving the joint $M$-best-diverse problem. While the joint approach is superior with respect to quality of its results, it is inferior to the sequential method [4] with respect to runtime. For the case of binary submodular energies, the approximate solver in [22] and the exact solver in [23] are several times slower than the sequential technique [4] for a normally sized image. Obviously, this is a major limitation when using it in a practical setting. Furthermore, the difference in runtime grows with the number $M$ of configurations.

In this work, we show that in case of binary submodular energies an *exact* solution to the joint $M$-best-diverse problem can be obtained significantly faster than the approximate one with the sequential method [4]. Moreover, the difference in runtime grows with the number $M$ of configurations.

**Related work**

The importance of the considered problem can be justified by the fact that a procedure of computing $M$-*best solutions* to discrete optimization problems was proposed over 40 years ago, in [28]. Later, more efficient specialized procedures were introduced for MAP-inference on a tree [34], junction-trees [30] and general graphical models [41, 13, 3]. However, such methods are however not suited for the scenario where *diversity* of the solutions is required, since they do not enforce it explicitly.

*Structural Determinant Point Processes* [27] is a tool for modelling probability distributions of structured models. Unfortunately, an efficient sampling procedure to obtain diverse configurations is feasible only for tree-structured graphical models. The recently proposed algorithm [8] to find $M$ *best modes* of a distribution is also limited to chains and junction chains of bounded width.

Training of $M$ *independent* graphical models to produce multiple diverse solutions was proposed in [15], and was further explored in [17, 9]. In contrast, we assume *a single fixed* model where configurations with low energy (hopefully) correspond to the desired output.

The work of [4] defines the $M$-best-diverse problem, and proposes a solver for it. However, the diversity of the solutions is defined sequentially, with respect to already extracted labelings. In contrast to [4], the work [22] defined the "joint $M$-best-diverse problem" as an optimization problem of a *single joint energy*. The most related work to ours is [23], where an efficient method for the joint $M$-best-diverse problem was proposed for submodular energies. The method is based on the fact that for submodular energies and a family of diversity measures (which includes e.g., Hamming distance) the set of M diverse solutions can be totally ordered with respect to the partial labeling order. In the binary labeling case, the $M$-best-diverse solutions form a nested set. However, although the method [23] is a considerably more efficient way to solve the problem, compared to the general algorithm proposed in [22], it is still considerably slower than the sequential method [4]. Furthermore, the runtime difference grows with the number $M$ of configurations.

Interestingly, the above-mentioned "nestedness property" is also fulfilled by minimizers of a parametric submodular minimization problem [12]. In particular, it holds for the monotonic max-flow method [25], which is also widely used for obtaining diverse labelings in practice [7, 20, 19]. Naturally, we would like to ask questions about the relationship of these two techniques, such as: "Do the joint $M$-best-diverse configurations form a subset of the configurations returned by a parametric submodular minimization problem?", and conversely "Can the parametric submodular minimization be used to (efficiently) produce the $M$-best-diverse configurations?" We give positive answers to both these questions.

**Contribution**

    • For binary submodular energies we provide a relationship between the joint $M$-best-diverse and the parametric submodular minimization problems. In case of "concave node-wise diversity measures" [1] we give *a closed-form formula* for the parameters values, which corresponds to the joint $M$-best-diverse labelings. The values can be computed in advance, prior to any optimization, which allows to obtain each labeling independently.

    • Our theoretical results suggest a number of efficient algorithms for the joint $M$-best-diverse

problem. We describe and experimentally evaluate the two simplest of them, sequential and parallel. Both are considerably faster than the popular technique [4] and are as easy to implement. We demonstrate the effectiveness of these algorithms on two publicly available datasets.

## 2 Background and Problem Definition

**Energy Minimization** Let $2^{\mathcal{A}}$ denote the powerset of a set $\mathcal{A}$. The pair $\mathcal{G} = (\mathcal{V}, F)$ is called a *hyper-graph* and has $\mathcal{V}$ as a finite *set of variable nodes* and $F \subseteq 2^{\mathcal{V}}$ as *a set of factors*. Each variable node $v \in \mathcal{V}$ is associated with a *variable* $y_v$ taking its values in a finite *set of labels* $L_v$. The set $L_{\mathcal{A}} = \prod_{v \in \mathcal{A}} L_v$ denotes the Cartesian product of sets of labels corresponding to a subset $\mathcal{A} \subseteq \mathcal{V}$ of variables. Functions $\theta_f \colon L_f \to \mathbb{R}$, associated with factors $f \in F$, are called *potentials* and define local costs on values of variables and their combinations. Potentials $\theta_f$ with $|f| = 1$ are called *unary*, with $|f| = 2$ *pairwise* and $|f| > 2$ *high-order*. Without loss of generality we will assume that there is a unary potential $\theta_v$ assigned to each variable $v \in \mathcal{V}$. This implies that $F = \mathcal{V} \cup \mathcal{F}$, where $\mathcal{F} = \{f \in F \colon |f| \geq 2\}$. For any non-unary factor $f \in \mathcal{F}$ the corresponding set of variables $\{y_v \colon v \in f\}$ will be denoted by $y_f$. *The energy minimization* problem consists in finding *a labeling* $\boldsymbol{y}^* = (y_v \colon v \in \mathcal{V}) \in L_{\mathcal{V}}$, which minimizes the total sum of corresponding potentials:

$$\arg \min_{\boldsymbol{y} \in L_{\mathcal{V}}} E(\boldsymbol{y}) = \arg \min_{\boldsymbol{y} \in L} \sum_{v \in \mathcal{V}} \theta_v(y_v) + \sum_{f \in \mathcal{F}} \theta_f(\boldsymbol{y}_f) \,. \tag{1}$$

Problem (1) is also known as *MAP-inference*. Labeling $\boldsymbol{y}^*$ satisfying (1) will be later called *a solution of the energy-minimization* or *MAP-inference problem*, shortly *MAP-labeling* or *MAP-solution*.

Unless otherwise specified, we will assume that $L_v = \{0, 1\}$, $v \in \mathcal{V}$, i.e. each variable may take only two values. Such energies will be called *binary*. We also assume that the logical operations $\leq$ and $\geq$ are defined in a natural way on the sets $L_v$. The case, when the energy $E$ decomposes into unary and pairwise potentials only, we will term as *pairwise case* or *pairwise energy*.

In the following, we use brackets to distinguish between upper index and power, i.e. $(\mathcal{A})^n$ means the $n$-th power of $\mathcal{A}$, whereas $n$ is an upper index in the expression $\mathcal{A}^n$. We will keep, however, the standard notation $\mathbb{R}^n$ for the $n$-dimensional vector space and skip the brackets if an upper index does not make mathematical sense such as in the expression $\{0, 1\}^{|\mathcal{V}|}$.

**Joint-DivMBest Problem** Instead of searching for a single labeling with the lowest energy, one might ask for a set of labelings with low energies, yet being significantly different from each other. In [22] it was proposed to infer such $M$ diverse labelings $\{\boldsymbol{y}^1, \ldots, \boldsymbol{y}^M\} \in (L)^M$ *jointly* by minimizing

$$E^M(\{\boldsymbol{y}\}) = \sum_{i=1}^{M} E(\boldsymbol{y}^i) - \lambda \Delta^M(\{\boldsymbol{y}\}) \tag{2}$$

w.r.t. $\{\boldsymbol{y}\} := \boldsymbol{y}^1, \ldots, \boldsymbol{y}^M$ for some $\lambda > 0$. Following [22] we use the notation $\{\boldsymbol{y}\}$ and $\{\boldsymbol{y}\}_v$ as shortcuts for $\boldsymbol{y}^1, \ldots, \boldsymbol{y}^M$ and $y_v^1, \ldots, y_v^M$ correspondingly. Function $\Delta^M$ defines diversity of arbitrary $M$ labelings, i.e. $\Delta^M(\{\boldsymbol{y}\})$ takes a large value if labelings $\{\boldsymbol{y}\}$ are in a certain sense diverse, and a small value otherwise.

In the following, we will refer to the problem (1) of minimizing the energy $E$ itself as to *the master problem* for (2).

**Node-Wise Diversity** In what follows we will consider only *node-wise diversity measures*, i.e. those which can be represented in the form

$$\Delta^M(\{\boldsymbol{y}\}) = \sum_{v \in \mathcal{V}} \Delta_v^M(\{\boldsymbol{y}\}_v) \tag{3}$$

for some *node diversity measure* $\Delta_v^M \colon \{0, 1\}^M \to \mathbb{R}$. Moreover, we will stick to *permutation invariant* diversity measures. In other words, such measures that $\Delta_v^M(\{\boldsymbol{y}\}_v) = \Delta_v^M(\pi(\{\boldsymbol{y}\}_v))$ for any permutation $\pi$ of variables $\{\boldsymbol{y}\}_v$.

Let the expression $[\![A]\!]$ be equal to 1 if $A$ is true and 0 otherwise. Let also $m_v^0 = \sum_{m=1}^{M} [\![y_v^m = 0]\!]$ count the number of 0's in $\{\boldsymbol{y}\}_v$. In the binary case $L_v = \{0, 1\}$, any permutation invariant measure can be represented as

$$\Delta_v^M(\{\boldsymbol{y}\}_v) = \bar{\Delta}_v^M(m_v^0) \,. \tag{4}$$

To keep notation simple, we will use $\Delta_v^M$ for both representations: $\Delta_v^M(\{\boldsymbol{y}\}_v)$ and $\bar{\Delta}_v^M(m_v^0)$.

**Example 1** (Hamming distance diversity). *Consider the common node diversity measure, the sum of Hamming distances between each pair of labels:*

$$\Delta_v^M(\{\boldsymbol{y}\}_v) = \sum_{i=1}^{M} \sum_{j=i+1}^{M} [\![y_v^i \neq y_v^j]\!]. \tag{5}$$

*This measure is permutation invariant. Therefore, it can be written as a function of the number $m_v^0$:*

$$\Delta_v^M(m_v^0) = m_v^0 \cdot (M - m_v^0). \tag{6}$$

**Minimization Techniques for Joint-DivMBest Problem** Direct minimization of (2) has so far been considered as a difficult problem even when the master problem (1) is easy to solve. We refer to [22] for a detailed investigation of using general MAP-inference solvers for (2). In this paragraph we briefly summarize existing *efficient* minimization approaches for (2).

As shown in [22] the sequential method DivMBest [4] can be seen as a greedy algorithm for approximate minimization of (2), by finding one solution after another. The sequential method [4] is used for diversity measures that can be represented by sum of diversity measures between each pair of solutions, i.e. $\Delta^M(\{\boldsymbol{y}\}) = \sum_{m_1=1}^{M} \sum_{m_2=m_1+1}^{M} \Delta^2(\boldsymbol{y}^{m_1}, \boldsymbol{y}^{m_2})$. For each $m = 1, \dots, M$ the method sequentially computes

$$\boldsymbol{y}^m = \underset{\boldsymbol{y} \in L_{\mathcal{V}}}{\arg\min} \left[ E(\boldsymbol{y}) - \lambda \sum_{i=1}^{m-1} \Delta^2(\boldsymbol{y}, \boldsymbol{y}^i) \right]. \tag{7}$$

In case of a node diversity measure (3), this algorithm requires sequentially solving $M$ energy minimization problems (1), with only modified unary potentials comparing to the master problem (1). It typically implies that an efficient solver for the master problem can also be used to obtain its diverse solutions.

In [23] an efficient approach for (2) was proposed for submodular energies $E$. An energy $E(y)$ is called submodular if for any two labelings $\boldsymbol{y}, \boldsymbol{y}' \in L_{\mathcal{V}}$ it holds

$$E(\boldsymbol{y} \vee \boldsymbol{y}') + E(\boldsymbol{y} \wedge \boldsymbol{y}') \leq E(\boldsymbol{y}) + E(\boldsymbol{y}'), \tag{8}$$

where $\boldsymbol{y} \vee \boldsymbol{y}'$ and $\boldsymbol{y} \wedge \boldsymbol{y}'$ denote correspondingly the node-wise maximum and minimum with respect to the natural order on the label set $L_v$.

In the following, we will use the term *the higher labeling*. The labeling $\boldsymbol{y}$ is *higher* than the labeling $\boldsymbol{y}'$ if $y_v \geq y'_v$ for all $v \in \mathcal{V}$. So, the labeling $\boldsymbol{y} \vee \boldsymbol{y}'$ is higher than $\boldsymbol{y}$ and $\boldsymbol{y}'$. Since the set of all labelings is a lattice w.r.t. the operation $\geq$, we will speak also about *the highest labeling*.

It was shown in [23] that for submodular energies, under certain technical conditions on the diversity measure $\Delta_v^M$ (see Lemma 2 in [23]), the problem (2) can be reformulated as a submodular energy minimization and, therefore, can be solved either exactly or approximately by efficient optimization techniques (e.g., by reduction to the min-cut/max-flow in the pairwise case). However, the size of the reformulated problem grows at least linearly with $M$ (quadratically in the case of the Hamming distance diversity (5)) and therefore even approximate algorithms require longer time than the DivMBest (7) method. Moreover, this difference in runtime grows with $M$.

The mentioned transformation of (2) into a submodular energy minimization problem is based on the theorem below, which plays a crucial role in obtaining the main results of this work. We first give a definition of the "nestedness property", which is also important for the rest of the paper.

**Definition 1.** *An $M$-tuple $(\boldsymbol{y}^1, \dots, \boldsymbol{y}^M) \in (L_{\mathcal{V}})^M$ is called* nested *if for each $v \in \mathcal{V}$ the inequality $y_v^i \leq y_v^j$ holds for $1 \leq i \leq j \leq M$, i.e. for $L_{\mathcal{V}} = \{0, 1\}$, $y_v^i = 1$ implies $y_v^j = 1$ for $j > i$.*

**Theorem 1.** *[Special case of Thm. 1 of [23]] For a binary submodular energy and a node-wise permutation invariant diversity, there exists a nested minimizer to the Joint-DivMBest objective* (2).

**Parametric submodular minimization** Let $\boldsymbol{\gamma} \in \mathbb{R}^{|\mathcal{V}|}$, $i = \{1, \dots, k\}$ be a *vector of parameters* with the coordinates indexed by the node index $v \in \mathcal{V}$. We define the *parametric energy minimization* as the problem of evaluating the function

$$\min_{\boldsymbol{y} \in L_{\mathcal{V}}} E^{\boldsymbol{\gamma}}(\boldsymbol{y}) := \min_{\boldsymbol{y} \in L} \left[ E(\boldsymbol{y}) + \sum_{v \in \mathcal{V}} \gamma_v y_v \right] \tag{9}$$

for all values of the parameter $\boldsymbol{\gamma} \in \Gamma \subseteq \mathbb{R}^{|\mathcal{V}|}$. The most important cases of the parametric energy minimization are

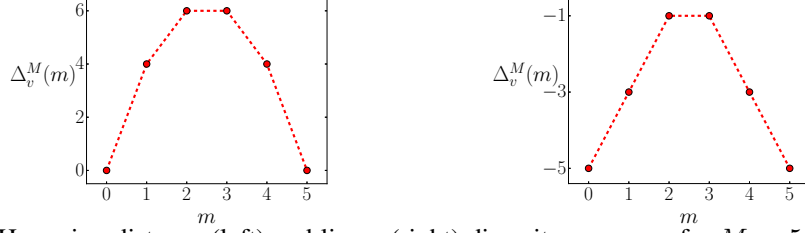

Figure 1: Hamming distance (left) and linear (right) diversity measures for $M = 5$. Value $m$ is defined as $\sum_{m=1}^{M} [\![ y_v^m = 0 ]\!]$. Both diversity measures are concave.

• the monotonic parametric max-flow problem [14, 18], which corresponds to the case when $E$ is a binary submodular pairwise energy and $\Gamma = \{\nu \in \mathbb{R}^{|\mathcal{V}|} : \nu_v = \gamma_v(\lambda)\}$ and functions $\gamma_v : \Lambda \to \mathbb{R}$ are non-increasing for $\Lambda \subseteq \mathbb{R}$.

• a subclass of the parametric submodular minimization [12, 2], where $E$ is submodular and $\Gamma = \{\gamma^1, \gamma^2, \ldots, \gamma^k \in \mathbb{R}^{|\mathcal{V}|} : \gamma^1 \geq \gamma^2 \geq \ldots \geq \gamma^k\}$, where operation $\geq$ is applied coordinate-wise.

It is known [38] that in these two cases, (i) the highest minimizers $\boldsymbol{y}^1, \ldots, \boldsymbol{y}^k \in L_\mathcal{V}$ of $E^{\gamma^i}$, $i = \{1, \ldots, k\}$ are nested and (ii) the parametric problem (9) is solvable efficiently by respective algorithms [14, 18, 12]. In the following, we will show that for a submodular energy $E$ the Joint-DivMBest problem (2) reduces to the parametric submodular minimization with the values $\gamma^1 \geq \gamma^2 \geq \ldots \geq \gamma^M \in \mathbb{R}^{|\mathcal{V}|}$ given in closed form.

## 3  Joint $M$-Best-Diverse Problem as a Parametric Submodular Minimization

Our results hold for the following subclass of the permutation invariant node-wise diversity measures:

**Definition 2.** *A node-wise diversity measure $\Delta_v^M(m)$ is called* concave *if for any $1 \leq i \leq j \leq M$ it holds*

$$\Delta_v^M(i) - \Delta_v^M(i - 1) \geq \Delta_v^M(j) - \Delta_v^M(j - 1). \tag{10}$$

There are a number of practically relevant concave diversity measures:

**Example 2.** *Hamming distance diversity* (6) *is* concave, *see Fig. 1 for illustration.*

**Example 3.** *Diversity measures of the form*

$$\Delta_v^M(m_v^0) = - \left( |m_v^0 - (M - m_v^0)| \right)^p = - \left( |2m_v^0 - M| \right)^p \tag{11}$$

*are concave for any $p \geq 1$. Here $M - m_v^0$ is the number of variables labeled as 1. Hence, $|m_v^0 - (M - m_v^0)|$ is an absolute value of the difference between the numbers of variables labeled as 0 and 1. It expresses the natural fact that a distribution of 0's and 1's is more diverse, when their amounts are similar.*

For $p = 1$ we call the measure (11) *linear*; for $p = 2$ the measure (11) coincides with the Hamming distance diversity (6). An illustration of these two cases is given in Fig. 1.

Our main theoretical result is given by the following theorem:

**Theorem 2.** *Let $E$ be binary submodular and $\Delta^M$ be a node-wise diversity measure with each component $\Delta_v^M$, $v \in V$, being permutation invariant and concave. Then a nested $M$-tuple $(\boldsymbol{y}^m)_{m=1}^M$ minimizing the Joint-DivMBest objective (2) can be found as the solutions of the following $M$ problems:*

$$\boldsymbol{y}^m = \arg\min_{\boldsymbol{y}_\mathcal{V}} \left[ E(\boldsymbol{y}) + \sum_{v \in \mathcal{V}} \gamma_v^m y_v \right], \tag{12}$$

*where $\gamma_v^m = \lambda \left( \Delta_v^M(m) - \Delta_v^M(m - 1) \right)$. In the case of multiple solutions in (12) the highest minimizer must be selected.*

We refer to the supplement for the proof of Theorem 2 and discuss its practical consequences below.

First note that the sequence $(\boldsymbol{\gamma}^m)_{m=1}^M$ is monotone due to concavity of $\Delta_v^M$. Each of the $M$ optimization problems (12) has the same size as the master problem (1) and differs from it by

unary potentials only. Theorem 2 implies that $\gamma^m$ in (12) satisfy the monotonicity condition: $\gamma^1 \geq \gamma^2 \geq \ldots \geq \gamma^M$. Therefore, equations (12) constitute the parametric submodular minimization problem as defined above, which reduces to the monotonic parametric max-flow problem for pairwise $E$. Let $\lfloor \cdot \rfloor$ denote the largest integer not exceeding an argument of the operation.

**Corollary 1.** *Let $\Delta_v^M$ in Theorem 2 be the Hamming distance diversity (6). Then it holds:*

*1. $\gamma_v^m = \lambda(M - 2m + 1)$.*

*2. The values $\gamma_v^m$, $m = 1, \ldots, M$ are symmetrically distributed around $0$:*

$$-\gamma_v^m = \gamma_v^{M+1-m} \geq 0, \text{ for } m \leq \lfloor (M+1)/2 \rfloor \quad \text{and} \quad \gamma_v^m = 0, \text{ if } m = (M+1)/2.$$

*3. Moreover, this distribution is uniform, that is $\gamma_v^{m+1} - \gamma_v^m = 2\lambda$, $m = 1, \ldots, M$.*

*4. When $M$ is odd, the MAP-solution (corresponding to $\gamma^{(M+1)/2} = 0$) is always among the $M$-best-diverse labelings minimizing (2).*

**Corollary 2.** *Implications 2 and 4 of Corollary 1 hold for any symmetrical concave $\Delta_v^M$, i.e., those where $\Delta_v^M(m) = \Delta_v^M(M + 1 - m)$ for $m \leq \lfloor (M+1)/2 \rfloor$.*

**Corollary 3.** *For* linear *diversity measure the value $\gamma_v^m$ in (12) is equal to $\lambda \cdot sgn\left(\frac{M}{2} - m\right)$, where $sgn(x)$ is a sign function, i.e., $sgn(x) = [\![x > 0]\!] - [\![x < 0]\!]$. Since all $\gamma_v^m$ for $m < \frac{M}{2}$ are the same, this diversity measure can give only up to $3$ different diverse labelings. Therefore, this diversity measure is not useful for $M > 3$, and can be seen as a limit of useful concave diversity measures.*

## 4 Efficient Algorithmic Solutions

Theorem 2 suggests several new computational methods for minimizing the Joint-DivMBest objective (2). All of them are more efficient than those proposed in [23]. Indeed, as we show experimentally in Section 5, they outperform even the sequential DivMBest method (7).

The simplest algorithm applies a MAP-inference solver to each of the $M$ problems (12) sequentially and independently. This algorithm has the same computational cost as DivMBest (7) since it also sequentially solves $M$ problems of the same size. However, already its slightly improved version, described below, performs faster than DivMBest (7).

**Sequential Algorithm** Theorem 2 states that solutions of (12) are nested. Therefore, from $y_v^{m-1} = 1$ it follows that $y_v^m = 1$ for labelings $\boldsymbol{y}^{m-1}$ and $\boldsymbol{y}^m$ obtained according to (12). This allows to reduce the size and computing time for each subsequent problem in the sequence.[2] Reusing the flow from the previous step gives an additional speedup. In fact, when applying a push relabel or pseudoflow algorithm in this fashion the total work complexity is asymptotically the same as of a single minimum cut [14, 18] of the master problem. In practice, this strategy is efficient with other min-cut solvers (without theoretical guarantees) as well. In our experiments we evaluated it with the dynamic augmenting path method [6, 24].

**Parallel Algorithm** The $M$ problems (12) are completely independent, and their highest minimizers recover the optimal $M$-tuple $(\mathbf{y^m})_{\mathbf{m}}$ according to Theorem 2. They can be solved fully in parallel or, using $p < M$ processors, in parallel groups of $M/p$ problems per processor, incrementally within each group. The overhead is only in copying data costs and sharing the memory bandwidth.

**Alternative approaches** One may suggest that for large $M$ it would be more efficient to solve the full parametric maxflow problem [18, 14] and then "read out" solutions corresponding to the desired values $\gamma^m$. However, the known algorithms [18, 14] would perform exactly the incremental computation described in the sequential approach above plus an extra work of identifying all breakpoints. This is only sensible when $M$ is larger than the number of breakpoints or the diversity measure is not known in advance (e.g., is itself parametric). Similarly, parametric submodular function minimization can be solved in the same worst case complexity [12] as non-parametric, but the algorithm is again incremental and would just perform less work when the parameters of interest are known in advance.

## 5 Experimental Evaluation

We base our experiments on two datasets: (i) The interactive foreground/background image segmentation dataset utilized in several papers [4, 31, 22, 23] for comparing diversity techniques; (ii) A new

Table 1: **Interactive segmentation**. The quality measure is a per-pixel accuracy of the best segmentation, out of $M$, averaged over all test images. The runtime is in milliseconds (ms). The quality for $M = 1$ is 91.57. `Parametric-parallel` is the fastest method followed by `Parametric-sequential`. Both achieve higher quality than `DivMBest`, and return the same solution as `Joint-DivMBest`.

|  | M=2 | | M=6 | | M=10 | |
|---|---|---|---|---|---|---|
|  | quality | time (ms) | quality | time (ms) | quality | time (ms) |
| `DivMBest` [4] | 93.16 | 2.6 | 95.02 | 11.6 | 95.16 | 15.4 |
| `Joint-DivMBest` [23] | **95.13** | 5.5 | **96.01** | 17.2 | **96.19** | 80.3 |
| `Parametric-sequential` (1 core) | **95.13** | 2.2 | **96.01** | 5.5 | **96.19** | 8.4 |
| `Parametric-parallel` (6 cores) | **95.13** | **1.9** | **96.01** | **4.3** | **96.19** | **6.2** |

dataset for foreground/background image segmentation with binary pairwise energies derived from the well-known PASCAL VOC 2012 dataset [11]. Energies of the master problem (1) in both cases are binary and pairwise, therefore we use their reduction [26] to the min-cut/max-flow problem to obtain solutions efficiently.

**Baselines** Our main competitor is the fastest known approach for inferring $M$ diverse solutions, the `DivMBest` method [4]. We made its efficient re-implementation using dynamic graph-cut [24]. We also compare our method with `Joint-DivMBest` [23], which provides an exact minimum of (2) as our method does.

**Diversity Measure** In all of our experiments we use the Hamming distance diversity measure (5). Note that in [31] more sophisticated diversity measures were used e.g., the *Hamming Ball*. However, the `DivMBest` method (7) with this measure requires to run a very time-consuming *HOP-MAP* [37] inference technique. Moreover, the experimental evaluation in [23] suggests that the exact minimum of (2) with Hamming distance diversity (5) outperforms `DivMBest` with a *Hamming Ball* distance diversity.

**Our Method** We evaluate both algorithms described in Section 4, i.e., sequential and parallel. We refer to them as `Parametric-sequential` and `Parametric-parallel` respectively. We utilize the dynamic graph-cut [24] technique for `Parametric-sequential`, which makes it comparable to our implementation of `DivMBest`. The max-flow solver of [6] is used for `Parametric-parallel` together with *OpenMP* directives. For the experiments we use a computer with 6 physical cores (12 virtual cores), and run `Parametric-parallel` with $M$ threads.

Parameters $\lambda$ (from (7) and (2)) were tuned via cross-validation for each algorithm and each experiment separately.

## 5.1 Interactive Segmentation

The basic idea is that after a user interaction, the system provides the user with $M$ diverse segmentations, instead of a single one. The user can then manually select the best one and add more user scribbles, if necessary. Following [4] we consider only the first iteration of such an interactive procedure, i.e., we consider user scribbles to be given and compare the sets of segmentations returned by the system.

The authors of [4] kindly provided us their 50 super-pixel graphical model instances. They are based on a subset of the PASCAL VOC 2010 [11] segmentation challenge with manually added scribbles. An instance has on average 3000 nodes. Pairwise potentials are given by contrast-sensitive Potts terms [5], which are submodular in the binary case. This implies that Theorem 2 is applicable.

**Quantitative comparison and runtime** of the different algorithms are presented in Table 1. As in [4], our quality measure is a per-pixel accuracy of the best solution for each test image, averaged over all test images. As expected, `Joint-DivMBest` and `Parametric-*` return the same, exact solution of (2). The measured runtime is also averaged over all test images. `Parametric-parallel` is the fastest method followed by `Parametric-sequential`. Note that on a computer with fewer cores, `Parametric-sequential` may even outperform `Parametric-parallel` because of the parallelization overheads.

## 5.2 Foreground/Background Segmentation

The Pascal VOC 2012 [11] segmentation dataset has 21 labels. We selected all those 451 images from the validation set for which the ground truth labeling has only two labels (background and one

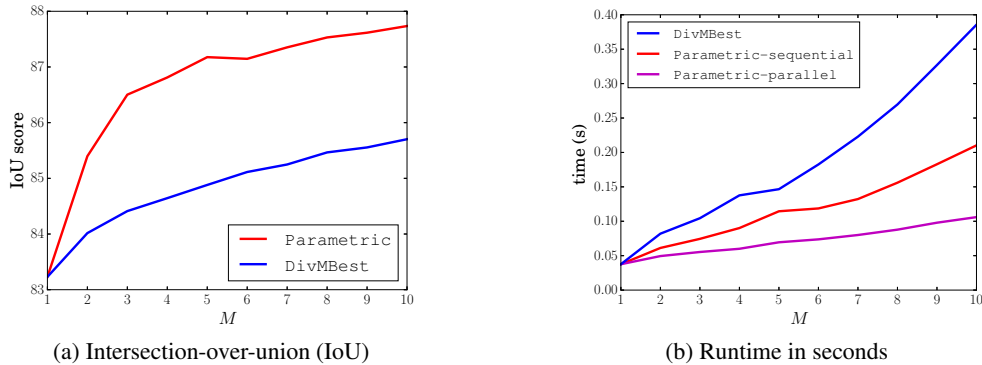

| | |
|:---:|:---:|
| (a) Intersection-over-union (IoU) | (b) Runtime in seconds |

Figure 2: **Foreground/background segmentation**. (a) Intersection-over-union (IoU) score for the best segmentation, out of $M$. `Parametric` represents a curve, which is the same for `Parametric-sequential`, `Parametric-parallel` and `Joint-DivMBest`, since they exactly solve the same Joint-DivMBest problem. (b) `DivMBest` uses dynamic graph-cut [24]. `Parametric-sequential` uses dynamic graph-cut and a reduced size graph for each consecutive labeling problem. `Parametric-parallel` solves $M$ problems in parallel using `OpenMP`.

of the 20 object classes) and which were *not* used for training. As unary potentials we use the output probabilities of the publicly available fully convolutional neural network `FCN-8s` [29], which is trained for the Pascal VOC 2012 challenge. This CNN gives unary terms for all 21 classes. For each image we pick only two classes: the background and the class-label that is presented in the ground truth. As pairwise potentials we use the contrastive-sensitive Potts terms [5] with a 4-connected grid structure.

**Quantitative Comparison and Runtime** As quality measure we use the standard Pascal VOC measure for semantic segmentation – average intersection-over-union (IoU) [11]. The unary potentials alone, i.e., output of `FCN-8s`, give 82.12 IoU. The single best labeling, returned by the MAP-inference problem, improves it to 83.23 IoU.

The comparisons with respect to runtime and accuracy of results are presented in Fig. 2a and 2b respectively. The increase in runtime with respect to $M$ for `Parametric-parallel` is due to parallelization overhead costs, which grow with $M$. `Parametric-parallel` is a clear winner in this experiment, both in terms of quality and runtime. `Parametric-sequential` is slower than `Parametric-parallel` but faster than `DivMBest`. The difference in runtime between these three algorithms grows with $M$.

## 6    Conclusion and Outlook

We have shown that the $M$ labelings, which constitute a solution to the Joint-DivMBest problem with binary submodular energies, and concave node-wise permutation invariant diversity measures can be computed in parallel, independently from each other, as solutions of the master energy minimization problem with modified unary costs. This allows to build solvers which run even faster than the approximate method of Batra et al. [4]. Furthermore, we have shown that such Joint-DivMBest problems reduce to the parametric submodular minimization. This shows a clear connection of these two practical approaches to obtaining diverse solutions to the energy minimization problem.

## Footnotes

[1]Precise definition is given in Sections 2 and 3.

[2] By applying "symmetric reasoning" for the label 0, further speed-ups can be achieved. However, we stick to the first variant in our experiments.

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
