[Supplementary Material]

# Supplementary Materials: Joint M-Best-Diverse Labelings as a Parametric Submodular Minimization

**Alexander Kirillov**[1]   **Alexander Shekhovtsov**[2]   **Carsten Rother**[1]   **Bogdan Savchynskyy**[1]

[1] TU Dresden, Dresden, Germany       [2] TU Graz, Graz, Austria

alexander.kirillov@tu-dresden.de

Below we provide two proofs of Theorem 2. The first one is restricted to pairwise energies. It is based on representing the submodular Joint-DivMBest problem (2) in the form of minimizing a convex multilabel energy. This problem is known as Convex MRF or as total variation (TV) regularized optimization with convex data terms. Thresholding theorems [8, 6, 5, 3, 9] then allow to break the problem into independent minimization and connect it to parametric mincut. This approach reveals an important link between our problem and the mentioned methods. It is also the shorter one. However, it is limited by the existing thresholding theorems and does not fully cover e.g., the higher order case (as discussed below).

The second proof is both more general (applies to arbitrary submodular energies) and simpler than the respective proves of the related results. It is self contained and uses only basic concepts of submodular function minimization, revealing the true simplicity of central fact that allows the problem to decouple.

## 7   Pairwise Case

For pairwise energies it holds $f = \{u, v\}$, $u, v \in \mathcal{V}$. Therefore, we will denote $\theta_f$ as $\theta_{u,v}$. The energy of the master problem (1) then reads

$$E(y) = \sum_{v \in \mathcal{V}} \theta_v(y_v) + \sum_{uv \in \mathcal{F}} \theta_{u,v}(y_u, y_v). \tag{13}$$

It is known [2] and straightforward to check that in the binary case it holds

$$E(y) = \text{const} + \sum_{v \in \mathcal{V}} a_v y_v + \sum_{uv \in \mathcal{F}} \Theta_{u,v}|y_u - y_v|, \tag{14}$$

where $a_v = \theta_v(1) - \theta(0)$ and $\Theta_{u,v} = \theta_{u,v}(0,1) + \theta_{u,v}(1,0) - \theta_{u,v}(0,0) - \theta_{u,v}(1,1)$. For submodular $E$, the values $\Theta_{u,v}$ are non-negative. In what follows, we will use the representation (14) and omit the constant in it, since it does not influence any further considerations.

A nested $M$-tuple $\{\boldsymbol{y}\} := (\boldsymbol{y}^1, \ldots, \boldsymbol{y}^M)$ is unambiguously specified by $|\mathcal{V}|$ numbers $m_v^0 \in \{0, \ldots, M\}$, $v \in \mathcal{V}$, where $m_v^0$ defines a number of labelings, which are assigned the label $0$ in the node $v$. The link between the two representations is given by

$$m_v^0 = \sum_m [\![y_v^m = 0]\!], \tag{15}$$

$$y_v^m = m \le m_v^0. \tag{16}$$

In other words, labelings $y^m$ are superlevel sets of $m^0 \colon \mathcal{V} \to \{0, \ldots, M\}$.

Let us write the Joint-DivMBest objective (2) as a function of $m^0$. The label $m \in \{0, \ldots, M\}$ denotes that exactly $m$ out of $M$ labelings in $\{\boldsymbol{y}\}$ are assigned the label $0$ in the node $v$. The unary cost assigned to a label $m$ in the node $v$ is equal to $a_v(M - m)$, since exactly $(M - m)$ labelings out of $M$ are assigned the label $1$ in the node $v$. The pairwise cost for a pair of labels $\{m, n\}$ in the

neighboring nodes $\{u, v\} \in \mathcal{F}$ is equal to $\Theta_{u,v}|m - n|$, since exactly $|m - n|$ labelings switch their label 0 to the label 1 between nodes $u$ and $v$. Therefore

$$\sum_{i=1}^{M} E(\boldsymbol{y}^i) = \sum_{v \in \mathcal{V}} a_v(M - m_v^0) + \sum_{uv \in \mathcal{F}} \Theta_{u,v}|m_u^0 - m_v^0|, \qquad (17)$$

where $m_v^0$ is defined as in (15).

Adding a node-wise diversity measure $\sum_{v \in \mathcal{V}} \lambda \Delta_v^M(\{\boldsymbol{y}\}_v) = \sum_{v \in \mathcal{V}} \lambda \Delta_v^M(m_v^0)$ and regrouping terms, one obtains that the Joint-DivMBest objective (2) is equivalent to

$$\sum_{v \in \mathcal{V}} \left( a_v(M - m_v^0) - \lambda \Delta_v^M(m_v^0) \right) + \sum_{uv \in \mathcal{F}} \Theta_{u,v}|m_u^0 - m_v^0| \qquad (18)$$

and must be minimized with respect to the labeling $\boldsymbol{m}^0 \in \{0, \dots, M\}^{\mathcal{V}}$.

Since the diversity measure $\lambda \Delta_v^M(m_v^0)$ is concave w.r.t. $m_v^0$, the unary factors $a_v(M - m_v^0) - \lambda \Delta_v^M(m_v^0)$ are convex. The pairwise factors $\Theta_{u,v}|m_u^0 - m_v^0|$ are also convex w.r.t. $m_u^0 - m_v^0$ due to non-negativity of $\Theta_{u,v}$.

For concave diversity the problem can be solved efficiently in time $O(T(n, m) + n \log M)$ [8], where $n = |\mathcal{V}|$, $m = |\mathcal{E}|$ and $T(n, m)$ is the complexity of a minimum $s$-$t$ cut procedure that can be implemented efficiently as parametric. Even for $m^0$ ranging in the continuous domain the complexity of the method [8] is polynomial, essentially matching the complexity of a single mincut. In particular, [8, Theorem 3.1] shows that a solution of such convex multilabel energy minimization problem decouples into $M$ problems of the form (12). Our Theorem 2 then follows. $\qquad \square$

## 7.1 Related Results and Limitations

For a general (not necessarily concave) permutation invariant nodewise diversity, the problem (17) can be still solved efficiently in time $O(nm \log \frac{n^2}{m} M^2 \log M)$ by either [10] or [1, 9]. On the other hand, the reformulation (14) expressing the regularizer as the function $|m_u^0 - m_v^0|$ holds for the pairwise case only. The results of Hochbaum [8] are as well limited to the pairwise case and involve min-cut / max-flow arguments.

Other related results are as follows. Chan and Esedoglu [5] give a thresholding theorem (related to Theorem 2) for the Rudin-Osher-Fatemi denoising model [11], Darbon and Sigelle [6] for TV-regularized $L_1$ and $L_2$ data fidelity problems, Chambolle [3] proposes further generalization towards thresholding of TV-like regularized convex problems in a finite dimensional space. The thresholding theorem of Chambolle [3] is applicable to higher order models, however the conditions on the regularizer are stricter than in Theorem 2: only a certain convex subclass of submodular functions qualifies.

# 8 General Case

In the following, we identify a simple and general thresholding theorem, applicable to arbitrary submodular (not only pairwise) functions. This constitutes a basis for a general proof of Theorem 2.

## 8.1 Nestedness

The set of labeling $L_{\mathcal{V}}$ together with the coordinate-wise maximum and minimum operations $\vee$, $\wedge$ forms a distributive lattice. The respective partial order $\mathbf{x} \leq \mathbf{y}$ is the coordinate-wise order $(\forall v \in \mathcal{V}) \, x_v \leq y_v$.

**Definition 3.** *Function $F \colon L_{\mathcal{V}} \to \mathbb{R}$ is* monotone *(resp.* antitone*) if for all $\boldsymbol{x} \leq \boldsymbol{y}$ there holds $F(\boldsymbol{x}) \leq F(\boldsymbol{y})$ (resp. $F(\boldsymbol{x}) \geq F(\boldsymbol{y})$).*

For example, a linear function $2^{\mathcal{V}} \to \mathbb{R} \colon \boldsymbol{y} \mapsto \sum_{v=1}^{\mathcal{V}} a_v y_v$ is monotone for $a_v \geq 0$; a multilabel modular function $\sum_{v=1}^{\mathcal{V}} \theta_v(y_v)$ is monotone if $\theta_v(y_v') \geq \theta_v(y_v)$ for all $y_v' \geq y_v$ and all $v \in \mathcal{V}$. The sum, minimum and maximum of monotone (resp. antitone) functions is monotone (resp. antitone). Note that, e.g., minimum of modular functions is in general not modular.

The following lemma provides sufficient conditions under which the parametric min-cut and the parametric submodular function minimization are monotone. It is essential for the subsequent derivation that we formulate it in a constructive form, i.e., not only existence of nested minimizers is shown but also the way to restore nestedness.

**Lemma 1** (See [12] Theorem 6.1). *Let $E\colon L_{\mathcal{V}} \to \mathbb{R}$ be submodular and $F\colon L_{\mathcal{V}} \to \mathbb{R}$ be antitone. Then for any minimizer $\boldsymbol{x}$ of $E$ and any minimizer $\boldsymbol{y}$ of $E + F$, the solution $\boldsymbol{y}' = \boldsymbol{y} \vee \boldsymbol{x}$ is a minimizer of $E + F$ and $\boldsymbol{x} \leq \boldsymbol{y}'$.*

*Proof.* Since $\boldsymbol{x}$ is a minimizer of $E$, $E(\boldsymbol{x}) \leq E(\boldsymbol{y} \wedge \boldsymbol{x})$. Adding this inequality to the submodularity inequality,

$$E(\boldsymbol{x} \vee \boldsymbol{y}) + E(\boldsymbol{x} \wedge \boldsymbol{y}) \leq E(\boldsymbol{x}) + E(\boldsymbol{y}), \tag{19}$$

we obtain

$$E(\boldsymbol{x} \vee \boldsymbol{y}) \leq E(\boldsymbol{y}). \tag{20}$$

For antitone $F$ and $\boldsymbol{x} \vee \boldsymbol{y} \geq \boldsymbol{y}$ we have $F(\boldsymbol{x} \vee \boldsymbol{y}) \leq F(\boldsymbol{y})$. Adding to (20), we get

$$(E + F)(\boldsymbol{x} \vee \boldsymbol{y}) \leq (E + F)(\boldsymbol{y}). \tag{21}$$

Since $\boldsymbol{y}$ was a minimizer of $E + F$, it follows that $\boldsymbol{x} \vee \boldsymbol{y}$ is a minimizer of $E + F$ as well. $\square$

It follows that the minimal minimizer of $E$ is nested (contained in the case of set functions) in the minimal minimizer of $E + F$. Symmetrically, if $E$ is submodular and $F$ is monotone, then $\boldsymbol{x} \wedge \boldsymbol{y} \leq \boldsymbol{y}$ is a minimizer of $E + F$. Similar results appear in [8], [4, Lemma 3.4] in a somewhat less general form. Fleischer and Iwata [7, Lemma 3.1] proves nestedness under a related condition called a *strong map*. It can be easily shown that for submodular $E + F$ the map $E \to E + F$ is a strong map iff $F$ is antitone.

## 8.2 Thresholding Theorem

Let a function $\mathcal{E}\colon (L_{\mathcal{V}})^M \to \mathbb{R}$ of a tuple $\{\boldsymbol{y}\}$ has the expression

$$\mathcal{E}(\{\boldsymbol{y}\}) = \sum_{m=1}^{M} E_m(\boldsymbol{y}^m), \tag{22}$$

for some functions $E_m\colon L_{\mathcal{V}} \to \mathbb{R}$. We will show that such a decomposition holds for the Joint-DivMBest objective $E^M$ (2) under conditions of Theorem 2 (cf. coarea formula in [3]). Consider minimizing the function $\mathcal{E}$ over *nested* tuples $\{\boldsymbol{y}\}$.

**Theorem 3** (Thresholding). *Let $E_m\colon L_{\mathcal{V}} \to \mathbb{R}$ be submodular for each $m$ and $(E_m - E_{m-1})$ be antitone for each $m > 1$. Then the joint problem,*

$$\min_{\{\boldsymbol{y}\}} \sum_{m=1}^{M} E_m(\boldsymbol{y}^m) \ \ s.t. \ \{\boldsymbol{y}\} \ is \ nested, \tag{23}$$

*decouples into independent problems*

$$\sum_m \min_{\boldsymbol{y} \in L_{\mathcal{V}}} E_m(\boldsymbol{y}). \tag{24}$$

Where "decouples" means that the optimal values of both problems are equal and there is a simple mapping between their optimal solutions.

*Proof.* We will prove the theorem by constructing, out of independent minimizers $\hat{\boldsymbol{y}}^m$, $m = 1, \dots M$, a nested tuple of independent minimizers $\{\bar{\boldsymbol{y}}\}$.

Assume that $\hat{\boldsymbol{y}}^k \not\geq \hat{\boldsymbol{y}}^m$ for $k > m$. The function $E_k$ can be expressed as

$$E_k = E_m + F, \tag{25}$$

where $F = \sum_{l=m+1}^{k}(E_l - E_{l-1})$ is antitone (by conditions of the theorem). We have: $\hat{\boldsymbol{y}}^m$ minimizes $E_m$, $\hat{\boldsymbol{y}}^k$ minimizes $E_k$. By the nestedness Lemma 1, it must be that $\hat{\boldsymbol{y}}^k \vee \hat{\boldsymbol{y}}^m$ also minimizes $E_k$. Moreover, $\hat{\boldsymbol{y}}^k \vee \hat{\boldsymbol{y}}^m \geq \hat{\boldsymbol{y}}^m$.

By going in $m$ in order from 1 to $M$ and replacing $\hat{\boldsymbol{y}}^{m+1}$ with $\hat{\boldsymbol{y}}^{m+1} \vee \hat{\boldsymbol{y}}^m$ we obtain a tuple which is nested and each modifications has preserved optimality to (24). Let $\{\bar{\boldsymbol{y}}\}$ be the resulting nested tuple. It is feasible to the joint problem (23) and optimal to decoupled problem (24). Since also (23) is lower bounded by (24), the tuple $\{\bar{\boldsymbol{y}}\}$ is optimal to (23). □

**Corollary 4.** *Let elements of the tuple $\{\hat{\boldsymbol{y}}\}$ be defined as the highest minimizers of $E_m$:*

$$\hat{\boldsymbol{y}}^m = \bigvee \underset{\boldsymbol{y} \in L_\mathcal{V}}{\arg\min}\, E_m(\boldsymbol{y}). \tag{26}$$

*Then under conditions of Theorem 3 the tuple $\{\hat{\boldsymbol{y}}\}$ is nested and optimal to (23).*

*Proof.* The highest minimizer in (26), i.e., the maximum of the set of optimal solutions exists since the set of minimizers to a submodular function on a lattice is a lattice itself [12]. Since for each $k > m$, (i) the replacement $\hat{\boldsymbol{y}}^k \vee \hat{\boldsymbol{y}}^m$ is a minimizer of $E_k(\boldsymbol{y})$ (with the same substantiation as in the proof of Theorem 3) and (ii) $\hat{\boldsymbol{y}}^k$ is the highest minimizer of $E_k$, it therefore holds $\hat{\boldsymbol{y}}^k \vee \hat{\boldsymbol{y}}^m \leq \hat{\boldsymbol{y}}^k$. This is however the case only when $\hat{\boldsymbol{y}}^k \geq \hat{\boldsymbol{y}}^m$. □

### 8.3 Application to Submodular Joint-DivMBest Problem

Let $L_\mathcal{V} = \{0,1\}^\mathcal{V}$ and the master energy $E$ be submodular. Then the Joint-DivMBest problem according to Theorem 1 has the form

$$\min_{\{\boldsymbol{y}\}} \sum_{m=1}^{M} E(\boldsymbol{y}^m) - \Delta^M(\{\boldsymbol{y}\}) \text{ s.t. } \{\boldsymbol{y}\} \text{ is nested.} \tag{27}$$

In order to apply Theorem 3, we need to express the objective of this problem in the form (23). Since the master energy $E(\boldsymbol{y}^m)$ is the same for all $m$, clearly $E - E \equiv 0$ is an antitone function of $\boldsymbol{y}$. It remains to express $\Delta^M(\{\boldsymbol{y}\})$ in the form (23). Recall that any permutation invariant diversity measure $\Delta_v^M(\{y_v\})$ expresses as a function $g_v(x)$, where $x = \sum_{m=1}^{M}[\![y_v^m = 0]\!]$. Any function $g_v \colon \{0, \ldots M\} \to \mathbb{R}$ can be expressed as a prefix (cumulative) sum:

$$g_v(x) = g_v(0) + \sum_{m=1}^{M} \gamma_v^m [\![m \leq x]\!] = g_v(0) + \sum_{m \leq x} \gamma_v^m, \tag{28}$$

where $\gamma_v^m = g_v(m) - g_v(m-1)$ for $m \in \{1, \ldots, M\}$.

For concave diveristy measures, the discrete derivatives $\gamma_v^m$ of $g_v$ are monotone non-increasing in $m$. For a nested tuple $\{\boldsymbol{y}\}$ we have $y_v^m = [\![m \leq x]\!]$, and it holds

$$\Delta^M(\{\boldsymbol{y}\}) = \sum_{v \in \mathcal{V}} g_v(0) + \sum_{m=1}^{M} F_m(\boldsymbol{y}^m) \tag{29}$$

for $F_m(\boldsymbol{y}^m) = \sum_{v \in \mathcal{V}} \gamma_v^m y_v^m$.

Since the constant term $\sum_{v \in \mathcal{V}} g_v(0)$ can be ignored we proved that $\Delta^M(\{\boldsymbol{y}\})$ decomposes as (22). It remains to show that $-(F_m(\boldsymbol{y}) - F_{m-1}(\boldsymbol{y}))$ is antitone to fulfill conditions of Theorem 3 w.r.t. $-\Delta^M(\{\boldsymbol{y}\})$.

Indeed, the difference $F_m(\boldsymbol{y}) - F_{m-1}(\boldsymbol{y})$ expresses as $\sum_{v \in \mathcal{V}} a_v y_v$ with $a_v = \gamma_v^m - \gamma_v^{m-1}$. Since $\gamma_v^m$ is non-decreasing w.r.t. $m$, it holds $a_v \geq 0$ and $F_m - F_{m-1}$ is monotone. It implies $-(F_m - F_{m-1})$ is antitone.

As a result, the Joint-DivMBest problem (27) has the expression satisfying conditions of Theorem 3. Thus Theorem 2 holds. □

The study of the question which multilabel diversity measures have expression as a sum with monotone differences is left for the future work.