[Reviews · NeurIPS 2016]

Reviewer 1

Summary

The paper studies the problem of jointly inferring several diverse labelings for a binary submodular energy in a graphical model (the M-best-diverse problem). The paper establishes an interesting relationship between the joint M-best-diverse problem and the parametric submodular minimization problem; the latter has been studied in the work of Fleischer and Iwata for general submodular functions. The authors empirically evaluate the proposed algorithm and show that it achieves very good performance compared to the prior approaches for the M-best-diverse problem.

Qualitative Assessment

The contributions of this paper are novel and interesting both from a theoretical and experimental point of view. The theoretical result shows an interesting connection between the joint M-best-diverse problem and parametric submodular minimization. The experimental results show very good empirical performance of the proposed algorithm compared to the previous two methods both in terms of quality and running time. The contributions are novel and will likely lead to future work. The paper is generally well written. == post-rebuttal answer== I have read the entire rebuttal.

Confidence in this Review

2-Confident (read it all; understood it all reasonably well)


Reviewer 2

Summary

This paper proposes a new algorithm to jointly compute M diverse solutions with the objective function being their total energy minus the total diversity (measured in pairwise Hamming distance). The main theoretical result (Theorem 2) shows that for binary submodular energy, the M solutions can be solved using a parametric energy function with M given parameter values. In practice, this implies a sequential algorithm using the parametric max-flow-min-cut algorithm and a parallel algorithm which solves M min-cut problems in parallel. These algorithms are asymptotically faster than previous method for the same problem [22] which needs to solve an energy optimization problem on a graph of much larger size (quadratic to M). Experiments on interactive segmentation and VOC 12 segmentation datasets shows that the proposed methods are much faster than previous ones.

Qualitative Assessment

This paper is technically very sound. The main theoretical result is surprising to me and is not trivial to prove. The result also makes the connection between diverse solution problem and parametric min-cut, both of which are very interesting and popular problem/method in vision-inspired structured prediction problem. If I understand correctly, the sequential and parallel algorithms are asymptotically the same. However, according to Fig 2(b), the gap between the sequential and the parallel algorithms does grow linearly to M. This is kind of surprising to me and is not well explained. I also found it odd that this paper does not have the complexity explicitly stated.

Confidence in this Review

2-Confident (read it all; understood it all reasonably well)


Reviewer 3

Summary

This manuscript identifies a relationship between diversity with binary submodular energies and the parametric submodular minimization. Based on this, the joint M-best diverse labelings for the permutation invariant node-wise diversity measures can be obtained by running a non-parametric submodular minimization solver for M different values of parameter $\gamma$ in parallel. Importantly, the values for $\gamma$ can be computed in a closed form prior to any optimization. These lead two simple but efficient algorithms for the joint M-best diverse problem, which lead to better performance over those considered competitors in terms of runtime and result quality.

Qualitative Assessment

- In the title, it would be better to make clear the scope of the paper, which is M-best diverse labelings for binary submodular energies. The current title can be a bit misleading for future readers. - Also in the abstract and the introduction, it would be better to specify that the proposed stuff is valid for the class of the permutation invariant node-wise diversity measures. - The inclusion of representative qualitative results can be interesting for a part of readers.

Confidence in this Review

2-Confident (read it all; understood it all reasonably well)


Reviewer 4

Summary

The authors describe an improved sequential and a parallel algorithm for the case joint diverse best M solutions problem for binary submodular energy functions and nodewise diversity functions that are permutation invariant and concave.

Qualitative Assessment

This work builds primarily on the work of Kirillov but feels a bit incremental in some ways as the main theoretical result is very reminiscent of previous work. The fact that it leads to a better algorthimic approach is important, I think that the comparison with previous work could be stated more clearly in the draft. I would have liked to see an actual complexity analysis of the algorithms. That is, the authors claim that it leads to an improvement that scales with M, but that doesn't seem to be presented theoretically. What is the computational complexity of the algorithms involved? Does this improve the theoretical asymptotic complexity, or is it just a constant time improvement? In the worst case I think that they may be the samThis kind of analysis should be possible to state for the simple pairwise binary models discussed in this work. The same goes for the parallel algorithm. The draft is difficult to read because of a number of typos. This can certainly be corrected with some effort. Also, it appears that the main theoretical contribution is imprecisely stated. This should be corrected as it makes it difficult to read/check the proofs. Miscellaneous typos/comments: - line 61, "and proposes a solver to it" -> "and proposes a solver for it" - line 66, "e.g. Hamming distance" -> "e.g., Hamming distance" - line 104, "i.e. each variable may take only" -> "i.e., each variable may take only", this typo occurs in many places throughout the draft. - line 135-136, "In this paragraph we briefly summarize existing efficient minimization approaches for (2)" -> this is the end of the paragraph so it doesn't make any sense. - line 142, "this algorithm requires to sequentially solve M" -> "this algorithm requires sequentially solving M" - line 161, "which plays a crucial role for obtaining the main results of this work" -> "which plays a crucial role in obtaining the main results of this work" - line 176, I'm not sure that the definition of \Gamma here makes sense given that \Gamma should be a subset of R^{|V|}. In fact, I think that there is a problem with the definition of the parametric problem in line 165 as "i" does not appear anywhere in the definition. - line 204, "monotonous" -> "monotone"

Confidence in this Review

2-Confident (read it all; understood it all reasonably well)


Reviewer 5

Summary

This paper relates the problem of optimizing the Joint-Diverse-M-Best Problem to minimizing a parametric submodular function. Based on this relation two new algorithms are presented: (i) solving potentially up to M problems in parallel, and (ii) by exploiting the fact that the solutions to the problem form a nested tuple, the problem can be solved efficiently in a sequential manner. Experiments on two datasets show an improvement in computational time on the baseline algorithm Joint-DivMBest while getting the same exact solution. While both proposed methods return the exact solutions the runtime is even faster than the greedy approximation algorithm DivMBest.

Qualitative Assessment

- The paper is well written and comprehensible. - The relationship between the joint M best diverse and parametric submodular minimization is an interesting theoretical contribution. This leads to new algorithms that scale well. Especially for concave node wise diversity measures the M problems become independent and can thus be efficiently be implemented in parallel. The original problem has proven to be useful in computer vision, therefore reducing the computational burden is great. - Node-wise permutation invariant diversity measures are considered. Requiring permutation invariance is reasonable because wrong solutions may exist which are just permutations of the best solutions. This does, however, implicitly assume that good solutions are centered around the best solutions. - For interactive segmentation only the best out of M solutions is evaluated, all other M-1 solutions are neglected. It would be interesting to see how accurate these other solutions are. Further Details: *In the main paper tuples are denoted with (y^1,y^2,...) -- in the supplementary material (e.g. line 412,480) tuples are expressed by {y} which can easily be confused with sets. *line 515: diveristy -> diversity *Definition 1: The nested property for binary labels is easier to grasp with additional explanations that are given in line 230-231.

Confidence in this Review

2-Confident (read it all; understood it all reasonably well)